

# Evolutionary dynamics and geographic dispersal of beta coronaviruses in African bats

Babatunde O. Motayo[1,2], Olukunle Oluwapamilerin Oluwasemowo[1] and Paul A. Akinduti[3]

[1] Department of Virology, College of Medicine, University of Ibadan, Ibadan, Oyo, Nigeria
[2] Department of Medical Microbiology and Parasitology, Federal Medical Center, Abeokuta, Nigeria, Abeokuta, Ogun, Nigeria
[3] Department of Biological Science, Covenant University, Otta, Ogun, Nigeria

Corresponding author
Babatunde O. Motayo,
babatundemotayo@yahoo.com

## ABSTRACT

Bats have been shown to serve as reservoir host of various viral agents including coronaviruses. They have also been associated with the novel coronavirus SARS-CoV-2. This has made them an all important agent for CoV evolution and transmission. Our objective in this study was to investigate the dispersal, phylogenomics and evolution of betacoronavirus (βCoV) among African bats. We retrieved sequence data from established databases such as GenBank and Virus Pathogen Resource, covering the partial RNA dependent RNA polymerase (RdRP) gene of bat coronaviruses from eight African, three Asian, five European, two South American countries and Australia. We analyzed for phylogeographic information relating to genetic diversity and evolutionary dynamics. Our study revealed that majority of the African strains fell within Norbecovirus subgenera, with an evolutionary rate of $1.301 \times 10^{-3}$, HPD ($1.064 \times 10^{-3}$–$1.434 \times 10^{-3}$) subs/site/year. The African strains diversified into three main subgenera, *Norbecovirus*, *Hibecovirus* and *Merbecovirus*. The time to most common recent ancestor for *Norbecovirus* strains was 1973, and 2007, for the African *Merbecovirus* strains. There was evidence of inter species transmission of *Norbecovirus* among bats in Cameroun and DRC. Phlylogeography showed that there were inter-continental spread of Bt-CoV from Europe, China and Hong Kong into Central and Southern Africa, highlighting the possibility of long distance transmission. Our study has elucidated the possible evolutionary origins of βCoV among African bats; we therefore advocate for broader studies of whole genome sequences of BtCoV to further understand the drivers for their emergence and zoonotic spillovers into human population.

## INTRODUCTION

Coronaviruses are a large group of enveloped, positive sense, single stranded RNA viruses belonging to the order Nidovirales and family Coronaviridea (*Masters & Perlman, 2013*). The subfamily coronavirinae contains four genera: *Alphacoronavirus*, *Betacoronavirus*, *Gammacoronavirus* and *Deltacoronavirus* (*Masters & Perlman, 2013*). The International Committee on Taxonomy of Viruses (ICTV) recently adopted additional changes to the

nomenclature of Coronaviruses to include the inclusion of subgenera replacing the elsewhile lineage classification system (*Siddell et al., 2019*). Under this new system the genus *Betacoronavirus* was further classified into five subgenera *Sabecovirus, Merbecovirus, Norbecovirus, Embecovirus* and *Hibecovirus* (*Siddell et al., 2019*). *Betacoronaviruses* generally infect animals such as mammals and birds, they are the causative agents of many pathogenic diseases such as transmissible gastroenteritis of swine (TGEV), infectious bronchitis virus (IBD), mouse hepatitis virus (MHV), and bovine coronavirus (BCoV) (*Saif, 2004*). Coronavirus have been reported as early as 1930 (*Masters & Perlman, 2013*) but the earliest report of human coronavirus was in the year 1960, where two strains namely hCoV229-E and hCoVOC43 were described (*Hamre & Procknow, 1966*; *McIntosh et al., 1967*). Generally *Betacoranvirus* have been observed to cause paucisymptomatic disease in man and are largely known to be zoonotic. It was not until after the advent of the severe acute respiratory syndrome (SARS) outbreak in Hong Kong and parts of China in 2003, that studies into the zoonotic origin of the incriminated pathogen SARS CoV revealed that the Chinese Rinolophid bats maintained a genetically related strain of the SARS CoV (*Lau et al., 2005*; *Li et al., 2005*). This finding sparked up interest in bat CoV research. Another coronavirus termed Middle East severe respiratory syndrome virus (MERS CoV) was reported in the Arabian peninsula in 2012 (*Zaki et al., 2012*). Genetically similar strains to the MERS CoV were also isolated from *Pipistellus, Tyloncteris* and *Neoromica* bats (*Annan et al., 2013*).

In Africa large scale surveillance studies have identified diverse strains of coronavirus (CoV) circulating among African bats from Kenya, Ghana and Nigeria (*Drexler, Corman & Drosten, 2014*). Studies have also provided evidence that the human coronavirus hCoV229E originated from African bat CoV (AfrBtCoV) (*Corman et al., 2015*). Also, the rich fauna and biodiversity in Africa has made it a hotspot for emerging viral diseases. It is also inhabited by a diversity of bats which have been identified to serve as a reservoir of high consequence zoonotic diseases such as Marburg hemorrhagic fever and Rabies (*Annan et al., 2013*). Recently, a novel coronavirus SARS CoV-2 was identified to be the cause of a pandemic which originally broke out in Wuhan, Hubei province, China (*Zhou et al., 2020*). Some studies have also postulated that the SARS CoV-2 probably spilled over into human population through a zoonotic event involving Chinese SARS-related BtCoV (*Zhou et al., 2020*; *Li et al., 2020*).

The recent evidence of African bats as a potential reservoir host for several *Betacoronaviruses* (βCoV) gave rise to the conceptualization of this study which aims to investigate the spatial dispersal, phylogenomics and evolution of βCoV among African bats.

## MATERIALS AND METHODS

### Data collection

We searched for and downloaded partial or complete gene coding regions for the RNA dependent RNA polymerase sequences (RdRP) of Afr-BtCoV from GeneBank and the Virus Pathogen resource database https://www.viprbrc.org/. The data set generated contained African bat βCoV from eight countries, namely Nigeria, Kenya, Ghana,

Cameroun, Democratic Republic of Congo (DRC), Rwanda, Madagascar and South Africa ($n$ = 94), BtCoV from China, Hong Kong and Philippines in Asia; and France, Spain, Netherland, Italy and Luxemburg in Europe ($n$ = 95). Mexico and Brazil ($n$ = 3), Australia ($n$ = 1) and reference African CoV OC43, CoVHKU1, MERSCoV, and alpha coronavirus from Africa ($n$ = 35). Information such as country of origin, Host species, and date of collection were combined with the sequence information for the purpose of accurate phylogenetic determination. The final data stets had information from seven African countries, four European countries and three Asian countries. All the data used in this study can be assessed in Table S1. Majority of the African BtCoV sequences were generated by nRT-PCR using primers targeting the 440 bp partial RdRP gene region (*DeSouza et al., 2007*) and Sanger sequencing. Full genome sequences of ZBCoV were generated by both Sanger and ultra high throughput sequencing (UHTP) sequencing (*Quan et al., 2010*).

**Phylogenetic analysis**

Sequences were aligned using clustal W version 2.1 using default settings, the final alignment was 400bp in length. Phylogenetic trees were constructed in MEGA 7.0 software www.megasoftwre.net using the maximum likelihood method with a general time reversible GTR with a gama distributed rate variation ($T_4$) and a p-distance model with 1,000 bootstrap resampling. The final trees were then visualized in FigTree (http://tree.bio.ed.ac.uk/software/figtree/).

**Discrete phylogeographic analysis**

Aligned sequences were analyzed for evidence of sufficient temporal clock signal using TempEst version 1.5 (*Rambaut et al., 2016*). The relationship between root-to-tip divergence and sampling dates supported the use of molecular clock analysis in this study. Phylogenetic trees were generated by Bayasian inference through Markov chain Monte Carlo (MCMC), implemented in BEAST version 1.10.4 (*Suchard et al., 2018*). We partitioned the coding genes into first+second and third codon positions and applied a separate Hasegawa-Kishino-Yano (HKY+G) substitution model with gamma-distributed rate heterogeneity among sites to each partition (*Hasegawa, Kishino & Yano, 1985*). Two clock models were initially evaluated strict and relaxed molecular clock, with four different tree priors, constant population size, exponential population size, Bayesian Skyride plot and Gausian Markov Random Field Skyride plot. Each selected model was run for an initial 30,000,000 states. Models were compared using Bayes factor with marginal likelihood estimated using the path sampling and stepping stone methods implemented in BEAST version 1.10.4 (*Suchard et al., 2018*). The relaxed clock with Gausian Markov Random Field Skyride plot (GMRF) coalescent prior was selected for the final analysis. The MCMC chain was set at 100,000,000 states with 10% as burn in. Results were visualized using Tracer version 1.8 (http://tree.bio.ed.ac.uk/software/tracer/), all effective sampling size ESS values were >200 indicating sufficient sampling. Bayesian skyride analysis was carried out to visualize the epidemic evolutionary history using Tracer v 1.8 (http://tree.bio.ed.ac.uk/software/tracer/). To reconstruct the ancestral-state phylogeographic transmission across countries and hosts, we used the discrete-trait model

**Table 1 Distribution of βCoV infected bat species acording to country and βCoV lineage assignment.**

| Name | Abbreviation | Distribution | CoV sub-lineage |
|------|-------------|--------------|-----------------|
| *Elidion hevium* | *E. hevium* | Nigeria (6), Kenya (6) | *Norbecovirus* |
| *Neormicia* | *Neo* | South Africa (27) | *Merbecovirus* |
| *Scotophilus leucogaster* | *S. leucogaster* | Cameroun (2) | *Norbecovirus* |
| *Epomophorus gambianus* | *E. gambianus* | Cameroun (4) | *Norbecovirus* |
| *Rousettus aegypticaus* | *R. aegyticaus* | Rwanda (3) | *Norbecovirus* |
| *Epomophorus labiatus* | *E. labiatus* | Rwanda (1) | *Norbecovirus* |
| *Micropteropus pusillus* | *M. pisillus* | Cameroun (9), DRC (12) | *Norbecovirus* |
| *Epomophorus franquenti* | *E. franquenti* | DRC (3), Cameroun (1) | *Norbecovirus* |
| *Rhinolophus sinicus* | *R. sinicus* | China (8), | *Sabecovirus/Merbecovirus* |
| *Rousettus leschenauti* | *R. leschnauti* | H kong (15) | *Norbecovirus* |
| *Pipisterllus kuhilii* | *P. kuhilii* | Italy (5) | *Merbecovirus* |
| *Hipposiderus* | | Ghana (2) | *Hibercovirus* |
| *Tyloncteris pachypus* | *T. pachypus* | China (3) | *Sabecovirus* |

implemented in BEAST version 1.10.4 (*Suchard et al., 2018*). The Bayesian stochastic search variable selection (BSSVS) approach (*Lemey et al., 2009*) was used to explore the most important historical dispersal routes for the spread BtCoV across their countries of origin, as well as the most probable host-species transition. The spatiotemporal viral diffusion was then visualized using the Spatial Phylogenetic Reconstruction of Evolutionary Dynamics SPREAD3 software (*Bielejec et al., 2016*).

## RESULTS

We analyzed βCoV sequences from eight African countries distributed among eight bat species as shown in Table 1. The most abundant bat species sampled in this study was *Micropteropus. pussilus*, and Cameroon had the highest distribution of bat species sampled in this study. This result does not necessarily represent the true picture of bat species diversity in Africa, as some countries lack sequence information for bats due lack of surveillance.

Phylogenetic analysis of Bt-βCoV sequences revealed a significant proportion of the African strains, isolated from fruit bats fell within the sub-genera Norbecovirus formerly known as lineage D consisting of strains from Cameroon, DRC, Kenya, Madagascar and Nigeria (Fig. 1). Root to tip divergence showed the data set had a positive temporal signal (Fig. S1) with the correlation coefficient = 0.0286 and $R^2$ = 0.0818. The edited version of the final alignment of all sequences is available in Supplemental File 2. The MCC tree of the Afr-βBtCoV strains shows clearly the two major sub-genera *Norbercovirus*, and *Merbecovirus* (Fig. 2). We also observed that majority of the *Sabecovirus* (formerly lineage B) BtβCoV were isolated in Europe, precisely France and Spain and also China (Fig. 3). The TMRCA for African *Norbecovirus* dating back to 1973, 95% HPD

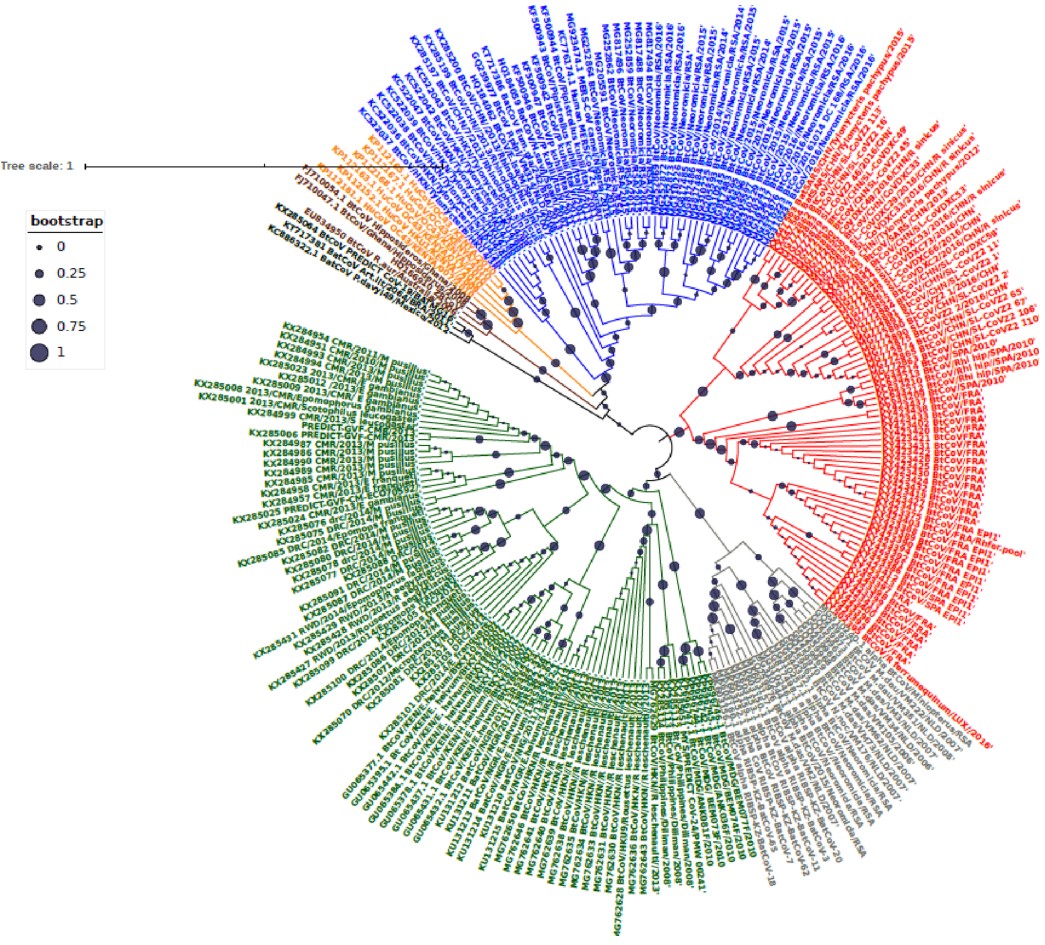

**Figure 1 Maximum Likelihood Phylogenetic tree, of Bat βCoV from this study.** The grey colored clade indicates alpha CoV from bats, green colored lines indicate lineage D (*Norbecovirus*) βCoV. The clade in red represents lineage B (*Sabecovirus*) Bat βCoV, while the blue colored clade represents lineage C (*Merbecovirus*) Bat βCoV. The orange colored labels represent Lineage E (*Embecovirus*), while the violet colored labels represent the *Hibercovirus* sub-genus. Bootstrap values are indicated as circles on the branches of the tree, the legend represents the value of the bootstrap where 1 is equivalent to 100%, 0.75 represents 75% 0.25 represents 25%.    

(1951–1990), and the TMRCA for *Merbecovirus* strains 2007 95% HPD (2003–2012). This shows that *Merbecovirus* is relatively recent and probably evolved from the existing *Norbecovirus* strains. Evolutionary rate of the African BtβCoV was set at $1.301 \times 10^{-3}$, HPD ($1.064 \times 10^{-3}$–$1.434 \times 10^{-3}$). This is slightly higher that the evolutionary rate for the ongoing SARS CoV-2 which has been estimated to have an evolutionary rate of $8.0 \times 10^{-4}$ (www.nextstrain.org/ncov/global).

Phylogeographic dispersal of the Bat β-CoV revealed numerous inter-continental spread events from China and Hong Kong into Central Africa (DRC and Kenya), Cameroun in West Africa, and South Africa, and also Mexico and Argentina in the Americas into West Africa Fig. 4. The AfrBtCoV strains displayed steady state population demography as depicted by their Bayesian Skyline plot (Fig. 5).

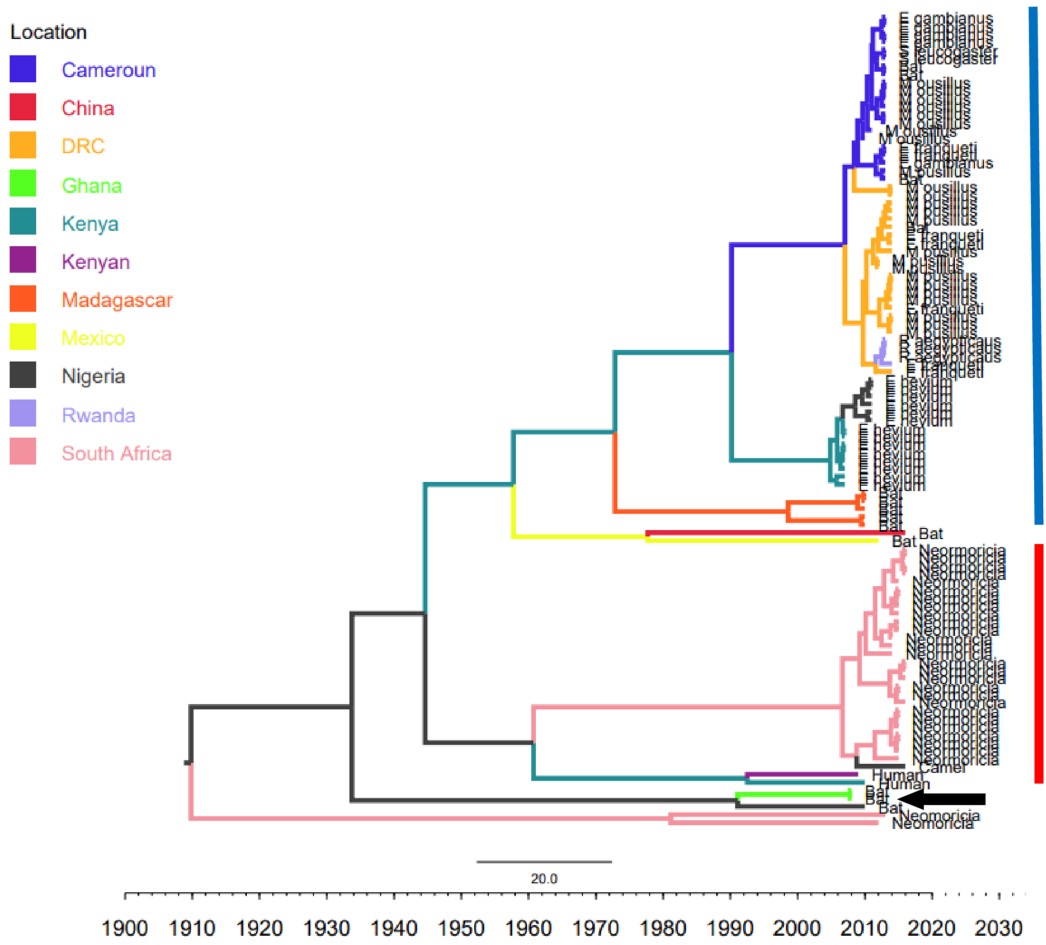

**Figure 2** **Time scaled Maximum clade credibility tree (MCC) of partial RdrP gene region of African Bt-βCoV.** Branch lengths are proportional to time in years, while branch colors are indicative of country of origin as shown in the legend. The blue horizontal bar represents Lineage D/*Norbecovirus* strains, while red represents Lineage C/*Merbecovirus*. The black arrow indicates *Hibercovirus* (formally under lineage B) strains.                                   

## DISCUSSION

### Phylogeny and evolutionary dynamics

Few studies have been carried in Africa on CoV among bats leaving a huge gap in epidemiologic information regarding BtβCoV in Africa.

Our report of *Norbecovirus* dominance among AfrBtCoV strains is in agreement with a previous report which identified the widespread circulation of *Norbecovirus* (Lineage D) among fruit bats in certain African regions (*Leopardi et al., 2016*). However, it was identified that isolates consisting largely of strains isolated among *Neomoricia* South African bats clustering within the sub-genus *Merbecovirus* (formerly Lineage C) together with strains isolated from Italy and Spain (Fig. 1). The phylogenetic classifications utilized in this study is based on the partial RdrP group unit (RGU), utilized for the rapid classification of field isolates of βCoV (*Tao et al., 2012*). The species-specific phylogenetic

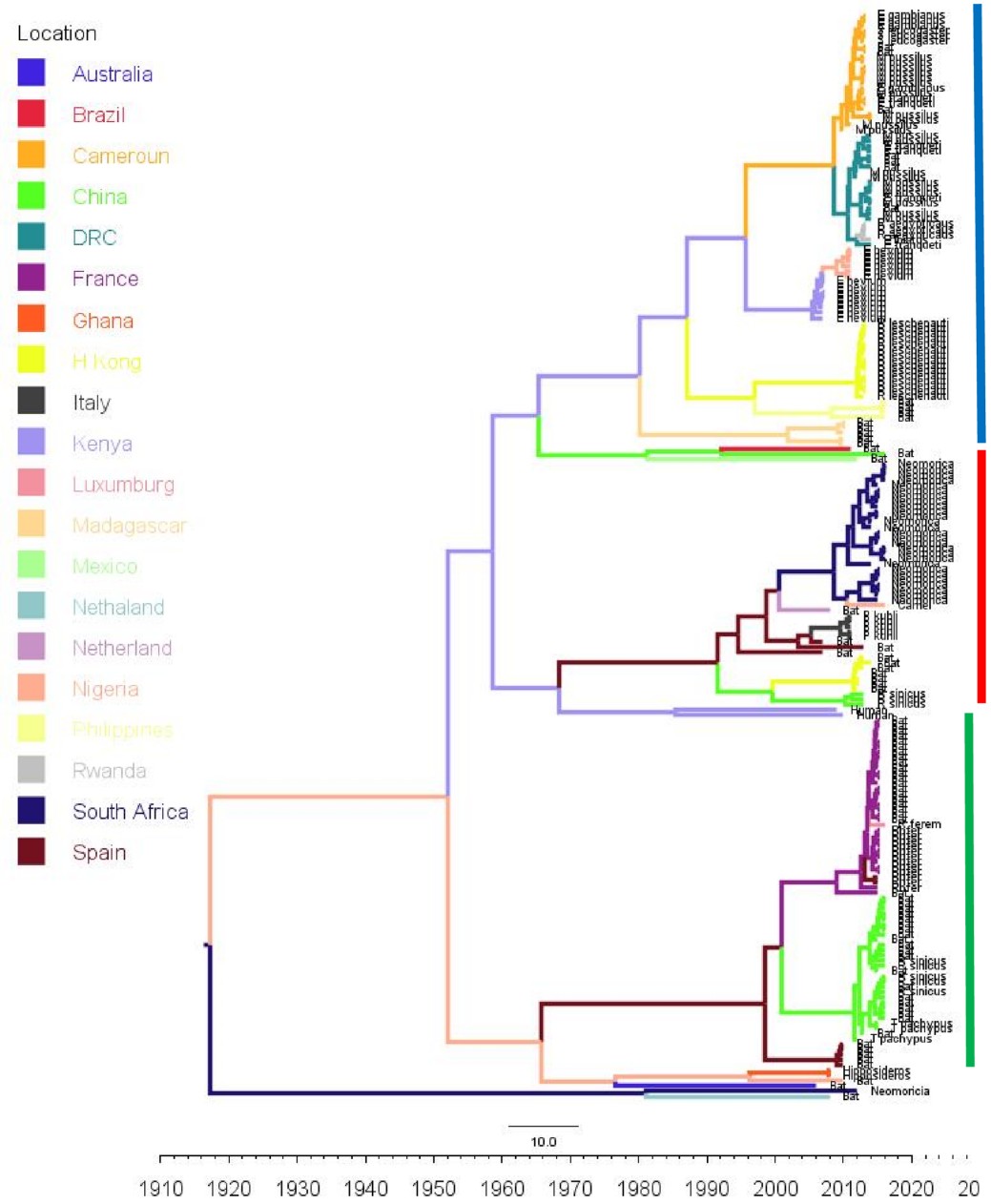

**Figure 3 Time scaled Maximum clade credibility tree (MCC) of partial RdrP gene region of Bt-βCoV from Africa and other global strains.** Time scaled Maximum clade credibility tree (MCC) with Bayesian phylogeographic reconstruction of partial RdrP gene region of Bt-βCoV. Branch lengths are proportional to time in years, while branch colors are indicative of country of origin as shown in the legend. Names of bat species of CoV isolation are shown in the tip labels. The blue horizontal bar represents Lineage D/*Norbecovirus* strains, green represents Lineage B/*Sabecovirus*, while red represents Lineage C/*Merbecovirus*.

clustering observed among the *Neomoricia* bats suggests limited inter-species βCoV transmission and host specific evolution among these species of bats in Africa as previously reported for BtCoV (*Wertheim et al., 2013*). Larger epidemiological studies are needed among these species of bats to shed more light as to the cause of this observed trend in

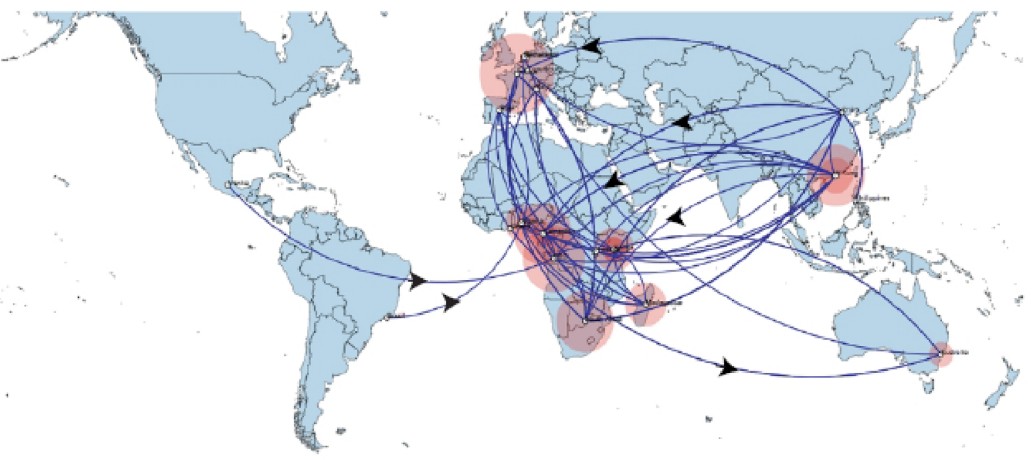

**Figure 4 Spaciotemporal diffusion of BtCoV, showing genetic spread of BtCoV across the world.**
Spaciotemporal diffusion of BtβCoV, the names of the countries of isolation are written in the map.
The diameter of the pink circles is proportional to the number of maximum clade credibility branches
(MCC). Concave lines (upward curving) show a clockwise spread movement between two points, while
the convex bending (downward curving) lines depict anticlockwise movement between the points.
The black arrows indicate the direction of the movements.

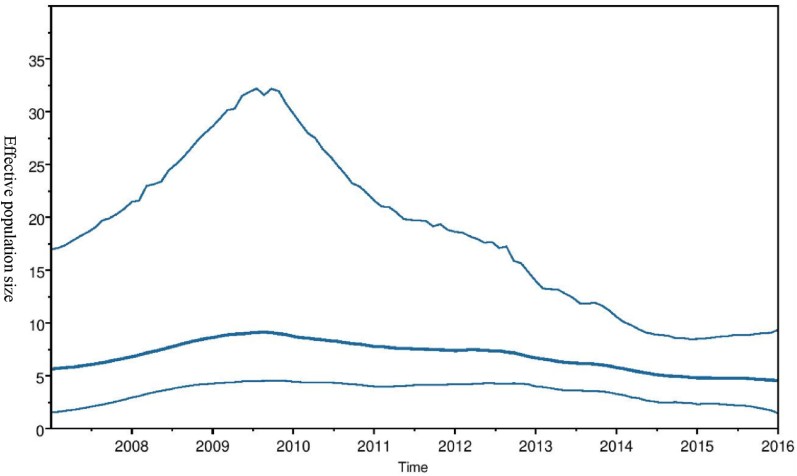

**Figure 5 Bayesian Skygrid plot of Afr Bt-βCoV.** Bayesian Skygrid plot showing effective virus popu-
lation size estimates over time of Afr Bt-βCoV. The solid blue line represents the median population size,
while the upper and lower blue lines represent the 95% high posterior density intervals (95% HPD) for
the population.                                         

Africa. Studies have shown that βCoV of subgenera *Merbecovirus* such as MERSCoV
are capable of both intra-species transmission and inter host transmission (*Min et al.,
2016*). In this study there seemed to be inter-species transmission among the *Norbercovirus*
(lineage D βCoV), evidenced by circulation of same *Norbecovirus* clade within different
species of bats from same country around with the same year of isolation. For instance
from Fig. 2 it can be seen that Cameroonian bat species *Micropteropus pussilus*,
*Epomophorus gambianus* and *Epomophorus franquenti*, were infected by the same

*Norbecovirus* clade, isolated around the year 2013. This is also observed among bat species from DRC (Fig. 2). This type of event allows for potential recombination and rapid evolution of this lineage, as previously reported (*Tao et al., 2017*). This type of observation was also reported in an earlier study of an inter-species transmission event of alpha CoV HKU10 in bat species of different orders (*Lau et al., 2012*).

Majority of the *Sabevovirus* strains reported in this study consisted of SARSr viruses from France, like EP11 strains which have been previously reported to be widely distributed across Europe and parts of Asia (*Ar Goulih et al., 2018*). The absence of *Sabecovirus* subgenera among the AfrBtCoV in our study seems to support the hypothesis that highly pathogenic CoV's such as SARS evolved outside the African continent. Although the lack of *Sabecovirus* in this study does not imply that this group of viruses is not currently circulating among African bats, as the closest subgenra *Hibecovirus* which was also formerly classified under Lineage B was identified in African Bats from Nigeria and Ghana (*Quan et al., 2010*), clustering with an Australian isolate ascension no: EU834950. This simply reflects the information gap in molecular data of BtCoV owing to poor surveillance in Africa. This also extends to information human coronaviruses such as hCoVOC43, hCoV229E, in which sequence data is limited to just a few countries such as Kenya and South Africa (*Sipulwa et al., 2016*; *Abidha et al., 2020*). Phylogenetically the genus *Rhinolophus* (Horse shoe bat) exhibited highest potential for intra-host diversity for BtCoV with the genus co-circulating both *Sabecovirus* and *Norbecovirus* strains Fig. 3. Our observation was similar to that of a study from Thailand (*Wacharapluesadee et al., 2015*) and supports the theory of diverse intra/inter-host transmission among different bat species which has been reported in previous studies (*Chu et al., 2008*; *Yuan et al., 2010*). Although we did not find this type of intra host genetic diversification among the African bat species in this study, it is believed that *Rhinolophus* bats are well distributed in Africa and are capable of zoonotic transmission of pathogenic hCoV such as SARS, as evidenced by a study that identified SARSrCoV antibodies among *Rhinolophus* bats in Africa (*Müller et al., 2007*).

The evolutionary rate reported in this study $1.301 \times 10^{-3}$ subs/site/year is slightly higher that the evolutionary rate for the ongoing SARS CoV-2 which has been estimated to have an evolutionary rate of $8.0 \times 10^{-4}$ (www.nextstrain.org/ncov/global). This is also slightly higher than the evolutionary rate reported for the partial RdRP gene of HuCoV OC43 of $1.06 \times 10^{-4}$ (*Vijgen et al., 2006*). A similar topology was also observed for the MCC tree which included βCoV from Asia and Europe, with a MRCA of 1915 (HPD 1880-1950) for all the strains, with the African strains showing the consistent TMRCA as described above (Fig. 3). The African *Norbecovirus* strains seemed to emerge from their parental strain at around the year 1960 (HPD, 1930–1970) this observation supports the hypothesis that *Norbecoviruses* could have been circulating in Africa long before they were first isolated. Whereas for the South African *Merbecovirus* (lineage C) strains seemed to emerge from their parental lineage around the year 1987 (Fig. 3), indicating a more recent introduction into Africa, however studies have dated their origin based on partial RdrP gene to as back as 1859 (*Lau et al., 2013*).

## Phylogeography of Bat B-Cov in Africa

The long distance spread events observed in our study such as the trans Atlantic and trans Mediterranean spread, may not necessarily represent actual transmission events, such as inter/intra-host transmission by migrating bat species, as bats have not been known to migrate across the Atlantic Ocean. However reports have shown the possibility of African bats to cover long distances during migration (*Ossa et al., 2012*). These observations simply represent genetic similarity and gene flow pathways of BtCoVs, which may due to other factors such as international trade in exotic and wild animals serving as intermediate host of these viruses. There were also dispersal of these viruses between France and some African countries such as Cameroun, South Africa and the DRC. There was also dispersal from Spain into Kenya, South Africa and Madagascar. We also observed dispersal across the Atlantic from Europe to Brazil, as well as across the Indian Ocean from Australia into East and Southern Africa. One limitation to this study is that we were unable to collect consistent data on the bat species from the reference isolates from other continents. Hence the data presented serves as a hypothetical model reflecting genetic dispersal of BtCoV and not species specific movements. The only dispersal event from Italy into Africa was into Nigeria. Studies have shown the potential for African fruit bats to migrate long distances covering thousands of Kilometers, for instance a study using satellite telemetry in Zambia showed that *Elodiun hevium* is capable of covering thousands of kilometers during migration (*Richter & Cumming, 2008*). Another study showed that African bats were capable of migration exceeding 2,000 Km (*Ossa et al., 2012*). Intra-continental dispersal events were observed between Cameroun, DRC and South Africa, as well as direct dispersal from Cameroon into Madagascar. The population demography reported in this study might not represent the true picture of the virus population, as the dataset utilized is limited by its size and might not represent the true demographic population of BtCoV in Africa.

## CONCLUSIONS

We have presented data on the phylodiversity and evolutionary dynamics of Afr-βCoV and their possible dispersal across the continent. Mutiple dispersal pathways were identified between Europe and East/Southern Africa; there were also evidence of spread of BtCoV strains from Asia into Africa. We also identified three CoV sub-genera (*Norbecovirus*, *Hibecovirus* and *Merbecovirus*) circulating among African bat species with the probability of inter-species transmission among bats. We also identified multiple corona virus sub-genera co-circulating in China among the bat specie *Rhinolophus sinicus*, with the capability of zoonotic transmission (*Yuan et al., 2010*; *Müller et al., 2007*). Study limitations include the lack of sufficient sequence data in GenBank covering AfrBtCoV, the relatively short genomic fragment analyzed and our inability to analyze spike protein sequence data of these viruses, as a result of paucity of African BtCoV spike protein sequences in established databases; this would have shed more light on their evolution in relation to infectivity and transmission. We have shown the importance of molecular surveillance of viruses with zoonotic potential such as coronaviruses. We advocate for broader trans-continental studies involving full genome sequences of

BtCoV to further understand the drivers for their emergence and zoonotic spillovers into human population.

### Funding
The authors received no funding for this work.

### Competing Interests
Babatunde O. Motayo has served as a reviewer for PeerJ.

### Author Contributions
- Babatunde O. Motayo conceived and designed the experiments, performed the experiments, analyzed the data, prepared figures and/or tables, authored or reviewed drafts of the paper, and approved the final draft.
- Olukunle Oluwapamilerin Oluwasemowo conceived and designed the experiments, performed the experiments, analyzed the data, authored or reviewed drafts of the paper, and approved the final draft.
- Paul A. Akinduti performed the experiments, authored or reviewed drafts of the paper, provided access key for internet enabled software, and approved the final draft.

### Data Availability
The raw data are available at Genbank. The 214 isolate accession numbers are listed in Table S2.

### Supplemental Information
Supplemental information for this article can be found online at http://dx.doi.org/10.7717/peerj.10434#supplemental-information.

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
