# Peer review of "Evolutionary dynamics and geographic dispersal of beta coronaviruses in African bats"

_PeerJ, doi:10.7717/peerj.10434_

## Round 0.1 · original submission · Major Revisions

As you see, all reviewers agree that while in general your contribution is significant, it has to be better presented. The problems relate to the overall logic of the exposition, the text itself, and the figures.

Reviewer 1 ·

Basic reporting

References are present. The Introduction is sufficient. The level of English is OK, although I found several typos:

Line 91-92: serve as reservoir -> serve as a reservoir
Line 105: codding -> coding
Line 107: viprbic.org -> viprbrc.org
Line 183: in earlier study -> in an earlier study

The main result of the article is a phylogenetic tree of coronaviruses from bats, based on the sequences of partial RNA dependent RNA polymerase. Unfortunately, the image of this tree (Figure 1) is low-resolution and I am unable to assess it. Aside from providing a better quality image, I would recommend adding a supplementary tree file, which could be opened by various tree-viewing programs. I like the authors idea to provide bootstrap values as branch width, however some sort of scale is required. It is unclear if the thickest width corresponds to the bootstrap value of 100%. Perhaps label one branch?

In the results and discussion section the authors state that: "We have however shown the importance of molecular surveillance of viruses with zoonotic potential such as coronaviruses". While I agree that such surveillance is of importance, I did not find a sufficient explanation in the article as to how this conclusion is derived from the results of the study or is related to them. Is it because there are so many gaps in data about the sequences of coronaviruses from bats in various regions?

I think Figure 2 and its implications should be better explained or omitted.

The text on Figure 4 is not readable.

I was not able to understand which lines are "concave" and which are "convex" on Figure 5. Perhaps there is some other way to show the paths of viral spread?

A population size estimate of 1E-1 (see Figure 6) is something I do not understand. What is the biological interpretation of such a population size? Is it possible?

Experimental design

The research is original. It would be beneficial to have a modern phylogenetic tree of bat coronaviruses including those from Africa in combination with biogeographic data. It can be stated that the research fills an identified knowledge gap.

The methods are reported in sufficient detail.

However, I think that some restrictions made in this study were not entirely justified.

1. The authors omitted countries with less than 5 sequenced bat coronaviruses. This probably excluded countries like Australia, where several (although apparently less that 5(?)) bat coronaviruses were sequenced. I think it would be interesting to know where do these viruses fit on the phylogenetic tree and how did they got to Australia. This is just one example. I don't see how focusing on a small number of countries (from the large amount inhabited by bats) makes the research better. I would prefer a more detailed project.

2. The phylogenetic tree was based on a single gene, the RNA dependent RNA polymerase. Why not use the complete genomes or sets of genes, which are probably available in most cases? This should be explained. Or is such data lacking?

Validity of the findings

As I already mentioned, some figures can not be assessed due to low resolution. Aside from this, I believe the stated methods seem adequate and have likely provide valid results.

I believe the study can be reproduced from the provided accession numbers and methods descriptions.

I suggest providing the alignments used for the phylogenetic analysis as a supplementary file.

The conclusions could be expanded. I felt that more could be said about why it is important to sequence bat coronavirus genomes and study how they spread from country to country. Perhaps the authors should also highlight the results that in their opinion have the most biological significance.

Additional comments

The work would be greatly improved if it expanded to a larger number of countries and used more genomic data (not just one gene).

·

Basic reporting

The paper is interesting, however, it should be more clearly written and be better structured.

The abstract is written in a very strange style: many abbreviations, methodological details, many numbers, many not sufficient/or poorly explained details - for example, "lineage D" - I think authors should remove all these details and focus on the most important biological results. For example, I would recommend writing more about dispersal and limited cross-species transmission. I interpret these findings as the key findings in the paper. If I am not right - authors should emphasise the most important findings. Currently, the paper has a very neutral style - just a description of some analyses.

The same recommendation is for the main text - there is not enough structurization of the paper.

FIGURES:
Quality of all figures is too poor - terminal branches of the trees are not readable.
line 254 - vertical instead of horizontal?
figure 1 - what is the root? How it was derived?
figure 6 - what is the Y-axis?

Line 117 - misprint

Experimental design

methods are written with all the necessary details.

Validity of the findings

Main findings - high dispersion of the viruses (probably together with bats) and limited cross-species transmission should be better demonstrated and written in more clear way.

·

Basic reporting

no comment

Experimental design

no comment

Validity of the findings

no comment

Additional comments

Review of “EVOLUTIONARY DYNAMICS AND GEOGRAPHIC DISPERSAL OF
BETA CORONAVIRUSES IN AFRICAN BATS” by Motayo et al.

The authors have put together a valuable new dataset, expanding our understanding of the diversity of betacoronaviruses. They focus on diversity in bats, mainly from Africa; this group is critical for our understanding of the potential of interspecies transmission of betacoronaviruses. While previous studies (e.g. Lu et al. Lancet 2020) have made use of complete genomes, this work uses partial or complete sequences of a single gene, RdRP; while providing lower phylogenetic resolution, it still provides valuable data on phylogeography and phylodynamics. In particular, they identify instances of inter-species transmission.

Overall, this manuscript is certainly timely and important. The main problem, I believe, is that the presentation of this data is unclear, and this impedes understanding. Overall, I found the manuscript difficult to read, some necessary details missing, and some statements mutually contradictory. I suggest a thorough revision of the text and figures for clarity. Furthermore, it is somewhat hard to understand what exactly the authors want us to take out of their work; a clearer summary of the key findings would be appreciated. I provide examples below.

1. According to the Data collection section of the Methods, the authors put together three distinct datasets; if I understand correctly, dataset 1 is nested within 2, and 2 within 3. However, these names are not used systematically, and I am confused by the interrelation between them. For example, the beginning of the Results section digs right into the details of dataset analysis, without pointing out which one is used.
2. If I understand correctly, dataset 1 is new, and is based on novel sequencing results. If so, information on sample acquision, sequencing protocols, assembly, etc. should be provided.
3. Many of the key results depend on the interpretation of Fig. 1. However, unfortunately the resolution of this figure is very poor, and I am unable to read the labels. Therefore, I am unable to see if the data supports claims like those in lines 175-179 and 180-182. The same holds for Fig. 4.
4. There are multiple typos, word omissions, etc.; e.g., “was” instead of “were” in L. 43, omitted space in L. 139, etc.
5. L112: what was the motivation for excluding countries with <5 sequences?
6. L121: Did you use CD-HIT to exclude some of the sequences? If yes, which sequences were excluded, and how many?
7. L180: what exactly is the evidence for inter-species transmission?
8. Phrases in L190-192 and 192-193 seem to contradict one another. The authors should clarify which explanation they favor, and why.
9. The dating of the lineage C based on one dataset is 2010 (95% HPD, 2006-2014, L196), but based on another, it’s 1983. Why the difference, which estimate do you favor, and why?
10. L190: “The absence of Lineage B” – you mean “in Africa”?

---

## Round 0.2 · Minor Revisions

There are some remaining editorial comments that need to be addressed.

Reviewer 1 ·

Basic reporting

The authors have improved their manuscript. The English is ok. Most data was shared, aside from the tree files. References are present. Results are relevent to hypotheses.

I still think that figures 1-4 should have improved resolution or better presentation, as most of them are difficult to read.

Figure 5 lacks a label on the Y axis.

The discription for Figure 4 now reads:

"Concave lines (upward curving) show a clockwise spread
movement between two points, while the convex bending (downward curving) lines depict anticlockwise movement between the points".

As I understand clockwise and anticlockwise movement, both kinds of lines correspond to left-to-right movement. I'm not sure if this is what the authors wanted to mean. Perhaps both upward curving and downward curving lines should be read clockwise? Then left-to-right movement would be shown with concave lines and right-to-left movement will be shown with convex lines. Or maybe I am confused. Why not use arrows?

Experimental design

No comment

Validity of the findings

No comment

---

## Round 0.3 · accepted · Accept

The remaining technical and editorial concerns have been resolved.